# Ncl1-mediated metabolic rewiring critical during metabolic stress

Ajay Bhat[1,2,*] , Rahul Chakraborty[1,2,*], Khushboo Adlakha[1], Ganesh Agam[1] , Kausik Chakraborty[1,2] , Shantanu Sengupta[1,2]

Nutritional limitation has been vastly studied; however, there is limited knowledge of how cells maintain homeostasis in excess nutrients. In this study, using yeast as a model system, we show that some amino acids are toxic at higher concentrations. With cysteine as a physiologically relevant example, we delineated the pathways/processes that are altered and those that are involved in survival in the presence of elevated levels of this amino acid. Using proteomics and metabolomics approach, we found that cysteine up-regulates proteins involved in amino acid metabolism, alters amino acid levels, and inhibits protein translation—events that are rescued by leucine supplementation. Through a comprehensive genetic screen, we show that leucine-mediated effect depends on a transfer RNA methyltransferase (NCL1), absence of which de-couples transcription and translation in the cell, inhibits the conversion of leucine to ketoisocaproate, and leads to tricarboxylic acid cycle block. We therefore propose a role of NCL1 in regulating metabolic homeostasis through translational control.

## Introduction

Cell requires optimal amount of nutrients for the synthesis of macromolecules like lipid, protein, and nucleotides. Activation of these anabolic processes and the repression of catabolic processes like autophagy promote cellular growth. However, when nutrients are limiting, anabolic processes are inhibited, and autophagy is activated (1). Among these processes, protein synthesis consumes a large portion of nutrients and energy (2, 3). Amino acids are the building block of proteins; therefore, its availability regulates the process of translation (4). Insufficient amino acid levels leads to the inhibition of protein synthesis (5) and induces the expression of genes required for synthesis of amino acids (6). Eukaryotic cells have two well-known pathways for sensing amino acids, which includes target of rapamycin complex 1 (TORC1) (7) and general

control nonderepressible 2 (Gcn2) (8). Limitation of amino acids promotes Gcn2-mediated signaling (9) and inhibits TORC1 (10), both of which lead to inhibition of protein synthesis (1, 11).

In contrast, there are reports indicating that excess of amino acids can lead to cellular toxicity (12, 13). Elevated levels of branch chain amino acids like leucine, isoleucine, and valine are associated with insulin resistance (14, 15) and are also responsible for neurological damage (16). Metabolic profiling has demonstrated that high levels of aromatic amino acids could be associated with the risk of developing diabetes (15, 17). Increased levels of phenylalanine because of either genetic mutation or exogenous supplementation lead to abnormal brain development (18, 19). Glutamate-mediated excitotoxicity leads to neuronal death (20); neonatal mice injected with high doses of glutamate shows increases in body fat and damaged hypothalamus region (21). Increased histidine intake in animals resulted in hyperlipidemia, hypercholesterolemia, and enlarged liver (12, 22). High levels of sulfur amino acids, cysteine, and methionine are reported to be toxic in various model systems (23, 24, 25, 26, 27). Supplementation of excessive methionine in rats suppresses food intake, diminishes growth, and also induces liver damage (12, 28). In mammals, excessive levels of cysteine have been demonstrated to be neurotoxic in many in vivo and in vitro studies (29, 30). Reports also suggest that elevated levels of cysteine may be associated with cardiovascular diseases (31, 32).

Thus, sensing of nutritional deprivation by the cell and cellular response because of nutrient-deprived conditions has been well studied. However, information on how cell responds and survives during nutritional excess is lacking. This information will help us understand how cellular homeostasis is maintained during accumulation of excess amino acid. This is important in the context of increased consumption of single amino acid as health supplements and their use as flavoring agents in recent years.

In this study, using Saccharomyces cerevisiae as a model, we used a combination of genetic, proteomic, transcriptomic and metabolic approach to understand cysteine-induced systemic alteration. We show that cysteine inhibits protein translation and alters the amino acid metabolism, effects of which are reversed by supplementation

[1]Council of Scientific and Industrial Research—Institute of Genomics and Integrative Biology, New Delhi, India   [2]Academy of Scientific and Innovative Research, Ghaziabad, India

Correspondence: kausik@igib.in; shantanus@igib.res.in
Ganesh Agam's present address is Department of Physical Chemistry, Ludwig-Maximilians-Universitat Munchen, Munich, Germany.
*Ajay Bhat and Rahul Chakraborty contributed equally to this work.

of leucine. We also show that a transfer RNA (tRNA) methyl-transferase (NCL1) is involved in survival during cysteine stress, absence of which inhibited the conversion of leucine to ketoi-socaproate (KIC)—a necessary step for mitigating the effect of cysteine. Thus, this study not only uncovers the cellular insights during accumulation of high levels of cysteine but also highlights the novel role of NCL1 in regulating the metabolism during excess cysteine.

## Results

### Metabolic alterations because of excess amino acids induce growth defect

Growth screening of *S. cerevisiae* (BY4741 grown in synthetic complete media [SC media]) in the presence of high concentrations of amino acids revealed that a few amino acids—cysteine, iso-leucine, valine, tryptophan, and phenylalanine—inhibited growth (Fig 1A). However, there was no change in the size of the cells when they were exposed to the toxic amino acids as compared with control and nontoxic amino acids (Fig S1A). Interestingly, we found

that isoleucine and phenylalanine had a "cidal" effect, whereas cysteine and tryptophan had "static" effect (Fig S1B). To test if the imbalance because of excess of an amino acid resulting in the growth inhibition could be abrogated in the presence of other amino acids, we did a comprehensive amino acid supplementation screening using 10-fold molar excess of the amino acids compared with that present in minimal media (Table S1). We found that the amino acid leucine could completely rescue the growth inhibition because of isoleucine, valine, tryptophan, and phenylalanine but not cysteine, where the rescue was partial (Fig 1B). The effect of leucine could be because the yeast strain used in this study, BY4741, is auxotrophic for leucine. To confirm this, we performed the same experiment using a prototrophic wild-type strain, S288C, and found that except cysteine, none of the other amino acids inhibited growth (Fig 1C). Furthermore, the growth inhibitory effect of cysteine was lower in this case compared with that observed in BY4741. We also found that there was no significant alteration in the growth of prototrophic yeast strain treated with excess cysteine in the presence or absence of leucine in the media (Fig S1C). This suggests that an optimum concentration of leucine is required to alleviate the toxicity of cysteine, which can be achieved by either in-tracellular synthesis or extracellular supplementation. This was

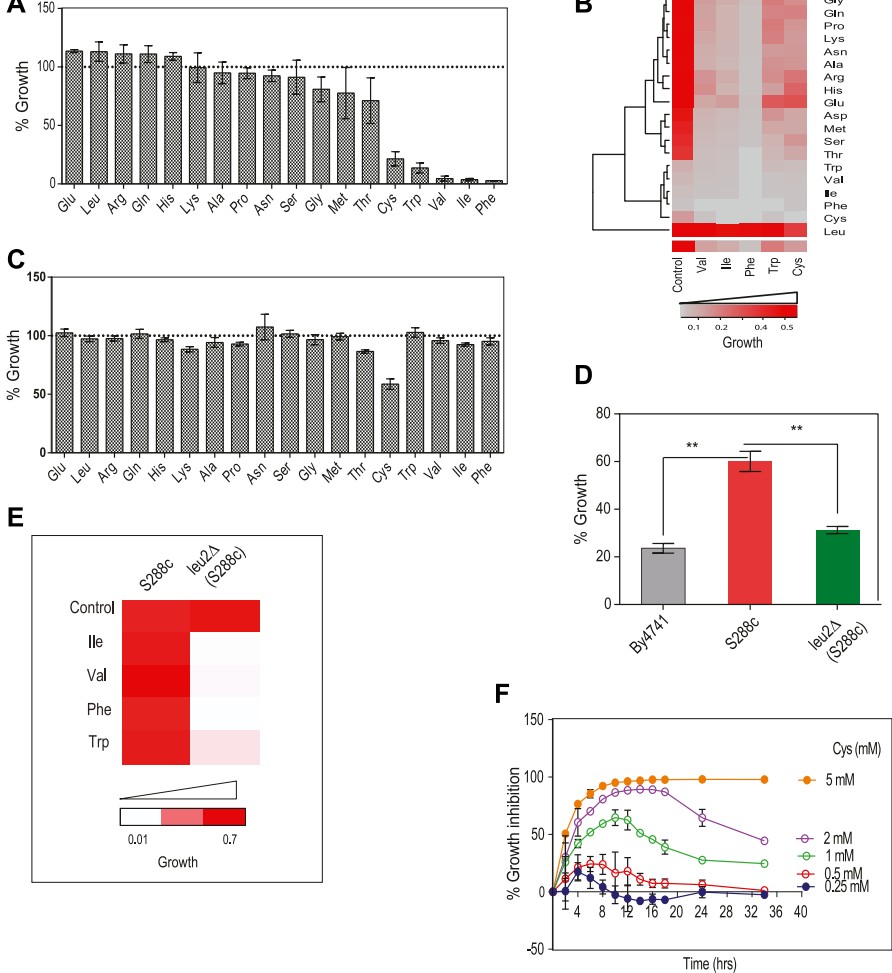

**Figure 1. Metabolic alterations because of excessive amino acids induce growth defect.**
**(A)** Cells of BY4741 strain were grown in higher concentration of amino acids (~10-fold molar excess than the amount present in the media) for 12 h, and growth was monitored by measuring OD at 600 nm. Percentage growth was then calculated with respect to the untreated cells (growth in SC media containing basal concentration of amino acids). Error bar represents SD (n = 3). **(B)** The higher concentration of toxic amino acids (Cys, Trp, Val, Ile, and Phe) were co-supplemented with higher levels (10 times as that of in media) of other amino acids, and growth was measured at 600 nm using multimode reader. The figure represents the growth matrix of cells grown with different combinations of amino acids. The bottom most row represents the growth of the cells during normal SC media (control) and treatment with high doses of single amino acids (Val, Ile, Phe, Trp, and Cys). Growth of the cells during these conditions co-treated with other amino acids (at higher concentrations) is represented in the above matrix. **(C)** Prototrophic strain (S288C) was grown with higher concentration of different amino acids for 12 h and then OD (at 600 nm) was measured to calculate the percentage growth with respect to untreated cells (growth in basal SC media). Error bar represents SD (n = 3). **(D)** Percentage of growth during cysteine treatment (1 mM) in BY4741, S288C, and leu2Δ (LEU2 deleted in S288C background) strains. Error bar represents SD (n = 3). Statistical significance was determined by unpaired *t* test (*$P < 0.05$, **$P < 0.01$, ***$P < 0.001$). **(E)** Growth of S288C and leu2Δ (LEU2 deleted in S288c background) strains in the presence of toxic amino acids (Ile, Val, Phe, and Trp). **(F)** Overnight grown cultures of yeast were reinoculated at an OD of 0.1 in SC media and growth was monitored in the presence (0.25–5 mM) and absence of ʟ-cysteine at different time points. The graph shows the kinetics of growth inhibition because of different concentrations of cysteine. Percentage growth inhibition was calculated by using the formula: {(OD of control cells − OD of cysteine-treated cells)/OD of control cell × 100}. Error bar represents SD (n = 3).

further confirmed by deleting the gene involved in leucine bio-synthesis (LEU2) in a prototrophic strain (S288C), which increased the sensitivity of this strain toward cysteine (Fig 1D) and other amino acids (Fig 1E). To check if pretreatment of cysteine or leucine had an effect similar to the co-treatment of these amino acids, we grew yeast cells (BY4741) in the presence of either cysteine or leucine for 6 h, washed the media, and treated the cells with cysteine, leucine, or a combination of cysteine and leucine. In both the conditions, leucine was able to rescue cysteine-induced toxicity (Fig S1D). Because cysteine was the only amino acid whose growth inhibitory effect could not be completely recovered by leucine, we focussed our attention in understanding the effect of cysteine in yeast.

Cysteine inhibits the growth of yeast (BY4741) in a dose-dependent manner (Fig S1E). Interestingly, for each concentration of cysteine (other than 5 mM, at which there was very minimal growth), the growth inhibition increased with time, reached maximum, and then decreased (Fig 1F). The recovery in the growth inhibition clearly indicates that cells have response mechanisms that lead to cellular adaptation.

## Amino acid metabolism and protein translation plays a vital role in cysteine toxicity

To understand the pathways altered because of cysteine treatment, we used an iTRAQ-based proteomics approach and identified proteins that were differentially expressed at 6 h (early time point) and 12 h (later time point) (Fig S2A and Tables S2 and S3). We found that proteins involved in amino acid metabolism were mostly up-regulated at both the time points (Fig 2A). Notably, in line with the genetic links to leucine metabolism, there was a strong induction in the enzymes linked to the synthesis of branched chain amino acids, including leucine, isoleucine, and valine. However, exogenous addition of leucine reverted the expression of these metabolic enzymes (at 6 h), suggesting that induction of these enzymes was the response toward cysteine-induced stress (Fig 2B).

We therefore determined the relative levels of amino acids in wild-type and cysteine-treated cells (at 12 h) and found that the levels of lysine was extremely high while that of threonine was extremely low in the presence of cysteine (Fig 2C). The levels of other amino acids were also altered albeit to a much lesser extent. Furthermore, cysteine-induced increase in lysine and arginine and decrease in threonine was completely reversed in the presence of leucine.

Proteomics data also revealed alteration in proteins involved in the translation machinery (Fig S2A), and because leucine is known to activate protein translation, we tested if cysteine toxicity is associated with translation arrest. For this, we performed 35S-methionine incorporation assay to analyze the effect of cysteine on de novo protein synthesis and found that cysteine leads to a significant reduction in the rate of incorporation of 35S-methionine compared with untreated cells (Fig 2D). To prove that this effect was not because of intracellular conversion of cysteine to methionine through homocysteine, we did the same experiment in met6Δ strain (which blocks the conversion of homocysteine to methionine). Even in this strain, cysteine inhibited the incorporation of 35S-methionine

(Fig S2B). Translation inhibition was also confirmed using polysome profiling, where we found that high levels of cysteine decrease the abundance of polysomes (Fig 2E). Co-treatment of leucine and cysteine drastically increase the levels of polysomes and thus maintains the active translational state of the cells. This clearly shows that cysteine-induced amino acid alterations and translational defect could be rescued by leucine treatment.

## Genetic interactors of leucine-mediated rescue

To better understand the mechanism of leucine-induced alleviation of cysteine toxicity, we performed a genome-wide mutant screen to identify genes required for survival in the presence of high levels of cysteine and the genetic interactors involved in leucine-mediated abrogation of cysteine-induced toxicity. For this, ~4,500 of nonessential deletion strains were screened. During this screen, deletion strains were grown in the 96-well format in SD media, in the presence and absence of 1 mM cysteine, and cell density was measured using multimode reader. Ratio of the cell density in cysteine-treated and control cells was used to monitor the effect of cysteine on the growth of the deletion strain. The ratio from the whole library was log transformed, and the strains with 1.5 times higher or lower ratio than median ratio were considered as extremes. Using this less stringent criterion, we found 163 strains that show sensitivity toward cysteine. Replicate screen experiments were then performed using these 163 strains that were found to be sensitive in the first screen. A deletion strain was considered sensitive toward cysteine if they were sensitive in all the three replicates tested and the average growth was at least lower than 20% compared with wild type in the presence of cysteine (with P-value ≤ 0.05). Using these stringent criteria, 50 deletion strains were found to be sensitive toward cysteine (Fig 3A and Table S4).

Deletion of genes involved in amino acid metabolism like THR4 (threonine synthase), THR1 (homoserine kinase), HOM3 (aspartate kinase), LEU3 (regulates branched chain amino acid synthesis), STP2 (regulates the expression of amino acid permeases), and YML082W (paralog of STR2) were among these sensitive strains, which further highlighted the significance of amino acid metabolism in the presence of high levels of cysteine.

We further checked the growth of all these sensitive strains in the presence of cysteine and leucine. A reversal of cysteine-induced toxicity was observed in almost all these deletion strains, except for three strains, yml082wΔ, ncl1Δ, and ctr1Δ (Fig 3B). YML082W is an uncharacterized gene, which is predicted to have carbon sulfur lyase activity, and is a paralog of cystathionine γ synthase (str2), which converts cysteine to cystathionine. CTR1 is a high-affinity copper transporter of plasma membrane (33) and has a human homolog SLC31A1 (34). NCL1, a SAM-dependent m5C methyltransferase, methylates cytosine to m5C at several positions in various tRNA (35, 36). Interestingly, a leucine tRNA (CAA) is the only tRNA where it methylates at the wobble position (35, 37). This coupled with the fact that cysteine-mediated amino acid alterations are reverted by leucine treatment led us to focus on the role of NCL1 in abrogating the growth inhibitory effect of cysteine.

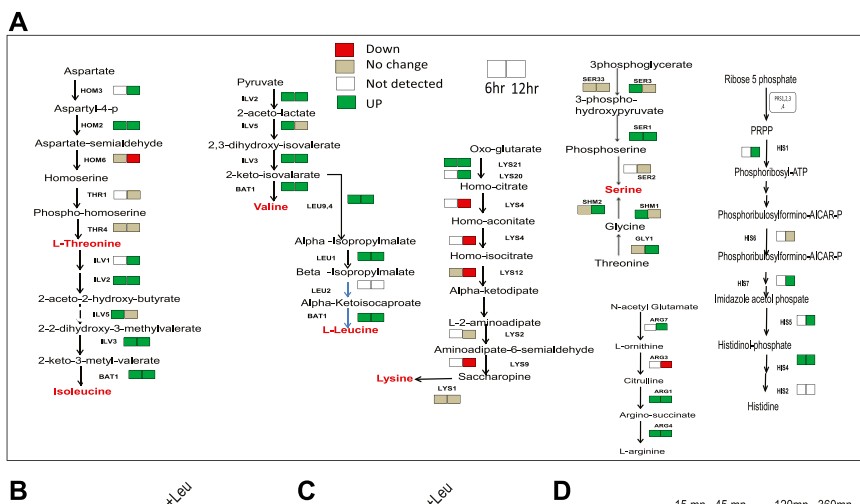

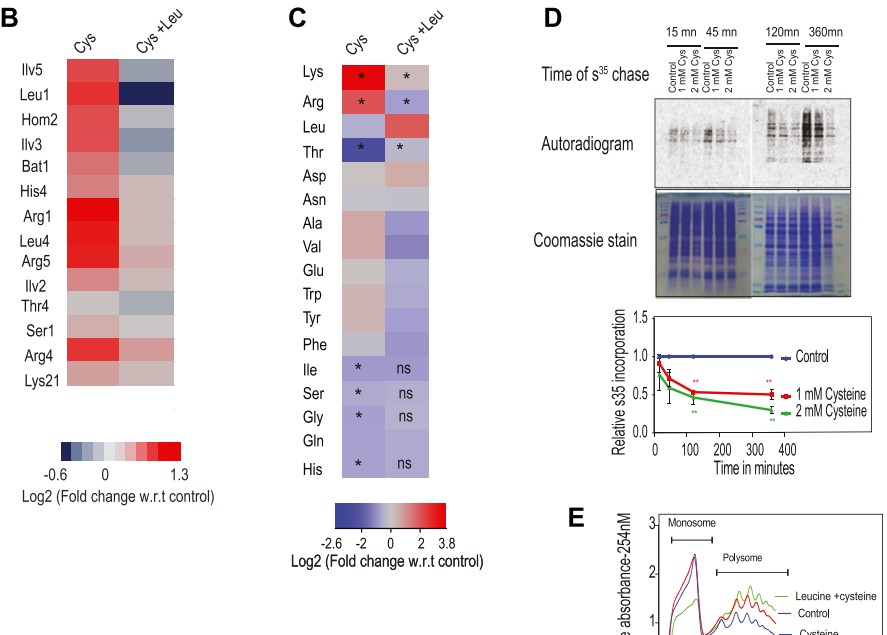

**Figure 2. Amino acid metabolism and protein translation play a vital role in cysteine toxicity.** **(A)** Cells were treated with cysteine for 6 and 12 h, and then iTRAQ-based relative quantification of proteins was performed. The figure represents the directionality of the expression of proteins involved in amino acid metabolism during 6 and 12 h of cysteine treatment. Left-side and right-side boxes for each protein denote its expression at 6 and 12 h of cysteine treatment, respectively. Red, green, and beige colored boxes represent the proteins down-regulated, up-regulated, and not changed, respectively, because of cysteine treatment. However, white colored box represents the proteins not detected in the proteomics experiment. **(B)** Heat map represents the relative expression of proteins (with respect to untreated condition) involved in amino acid metabolism during cysteine treatment and their status during co-treatment of leucine and cysteine. **(C)** Heat map represents the log fold change of different amino acids during cysteine (Cys) treatment and combination of leucine and cysteine (Leu + Cys). Statistical significance was determined by *t* test (n = 5, and n = 2 for Gln, Asn, and Trp, respectively). In the first column, the amino acids significantly altered by cysteine are labeled by "*" (*P* < 0.05); however, if the levels of these amino acids are significantly reverted by co-treatment of leucine and cysteine, then in the second column they are labeled by "*" (*P* < 0.05) or otherwise "ns" (nonsignificant). **(D)** 35S Methionine incorporation kinetics in the presence of different concentrations (1 and 2 mM) of cysteine for different time points. In the upper panel, the upper blot is for autoradiogram, which signifies S³⁵-Met incorporation in proteins, and the lower blot represents Coomassie stain of the same proteins, which represents equal loading of proteins. The lower panel is the quantification of 35S incorporation with respect to untreated cells. "**" indicates *P* < 0.01 (*t* test). Error bar represents SEM (n = 2). **(E)** Polysome profiling representing monosome and polysome fractions in the presence and absence of cysteine treatment and during co-treatment of leucine and cysteine.

## Leucine reprograms proteome through NCl1 to recover cysteine-mediated toxicity

We performed growth kinetics to confirm the role of NCL1 during cysteine-induced growth inhibition and leucine-mediated recovery. Deletion of NCl1 makes the cells incapable of abrogating cysteine-induced toxicity even in the presence of leucine (Fig 4A). To confirm the NCL1-dependent effect of leucine, we analyzed the proteomic profile of Wt and ncl1Δ strains in the presence of cysteine alone and a combination of cysteine and leucine. Although cysteine treatment in Wt cells leads to the up-regulation of proteins involved in amino acid metabolism both at 6 and 12 h and translation machinery at 12 h, it leads to down-regulation of proteins involved in translational machinery and amino acid metabolism in ncl1Δ strain (Fig S3A and Tables S5 and S6). This indicates that NCL1 may play a central role in regulating translation and metabolic rewiring

that is required to alleviate cysteine toxicity. In Wt cells, cysteine-induced differentially expressed proteins were reverted by leucine treatment (Figs 4B and S3B). The extent of cysteine-induced up-regulation of protein expression was considerably lower in ncl1Δ compared with Wt cells, and addition of leucine had no effect on their expression. This indicates global down-regulation of cysteine-induced response at the protein level in ncl1Δ, which could not be reverted even with leucine treatment. To ascertain if the changes induced by NCl1 in the presence of cysteine is at the transcriptional or translational level, we quantified changes in mRNA and protein levels during cysteine stress in Wt and ncl1Δ cells (Table S7). In Wt, we observed correlated changes in mRNA and protein expression upon cysteine treatment (*ρ* = 0.3, *P* < 0.0001, Fig 4C, top panel); however, no such correlation was found in Δncl1 strain (*ρ* = 0.07, *P* = 0.18, Fig 4C, bottom panel). This indicates a decoupling of translation and transcription in Δncl1 strain during cysteine treatment.

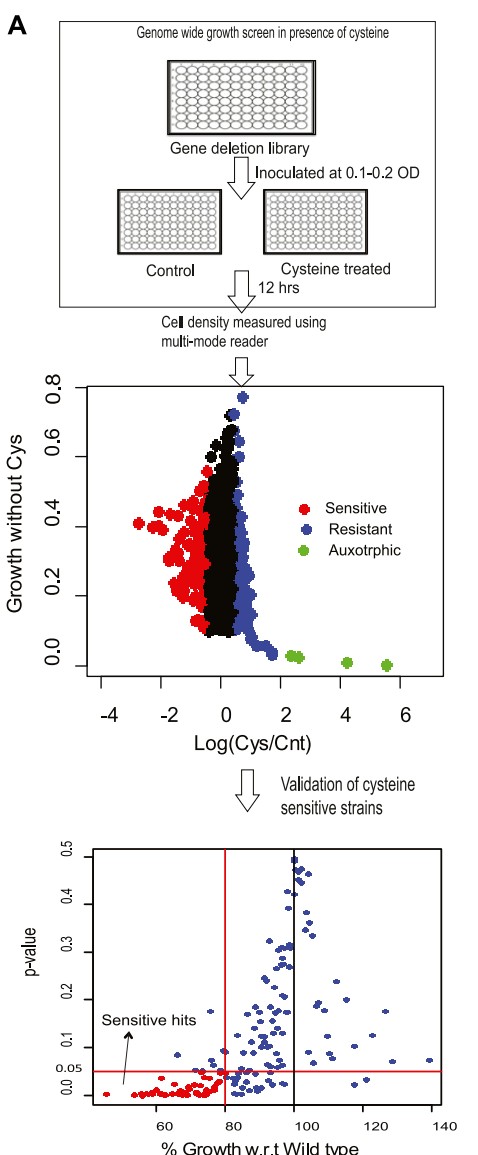

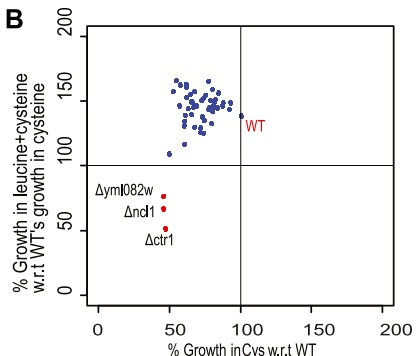

**Figure 3. Genetic interactors of leucine-mediated rescue.**
**(A)** Whole genome screen in the presence of 1 mM cysteine. The upper most panel represents the schematic of the genome-wide screen. The middle panel represents scatter plot between the logarithmic ratio of growth in the presence and absence of cysteine versus growth in basal media (SC media), for each deletion strain. In this panel, the deletion strains, which show growth sensitivity and resistance for cysteine, are symbolized with red and blue dots, respectively. Strains that were very slow growing and show growth advantage with cysteine are denoted by green dots. The lower most panel represents the validation of the cysteine-sensitive strains. Strains that show growth sensitivity toward high levels of cysteine in first round of whole genome screen were again grown in the presence of cysteine and growth was monitored and compared with wild-type (Wt) strain. Three biological replicates were performed, and average percentage growth of each strain with respect to Wt was calculated. Deletion strains whose growth was significantly different from Wt (*P*-value ≤ 0.05) (statistical significance was determined by unpaired *t* test), and grows at least 20% less than Wt, were considered to be sensitive for cysteine and are denoted by red dots. **(B)** Screening to identify genes required by leucine for alleviating cysteine-induced toxicity. Cysteine-sensitive deletion strains were grown with and without higher concentration of cysteine (1 mM), leucine (5 mM), or combination of leucine and cysteine. The figure represents the scatter plot between percentage growth during cysteine treatment versus percentage growth during co-treatment with leucine and cysteine, with respect to wild type's growth during cysteine treatment. Deletion strains in the left quadrant, which are denoted by red dots, represent the strains that are sensitive to cysteine but are not recovered by leucine.

To identify the pathways that could potentially be affected by NCL1 deletion, we analyzed the proteins whose expression was altered in cysteine and reverted by leucine in Wt but not in NCL1 deleted cells. These proteins belong to amino acid metabolism, specially branched chain amino acid biosynthetic proteins and oxidation–reduction process (Fig 4D). This indicates that NCL1 might be necessary for leucine to abrogate the amino acid imbalance induced by cysteine.

### Metabolism of leucine via NCL1 plays a vital role during cysteine-induced stress

To understand the role of NCL1 at the metabolic level, we measured the amino acids in ncl1Δ cells during cysteine treatment and co-treatment of cysteine and leucine. Interestingly, in ncl1Δ cells, the amino acid profile of cysteine-treated cells was similar to cells co-treated with cysteine and leucine as evident from their strong

correlation ($r^2$ = 0.99, *P*-value = 6.2 × 10$^{-10}$, Fig 5A). This further suggests that leucine-mediated metabolic rewiring depends on NCL1. Interestingly, we found a significant accumulation of leucine in ncl1Δ cells compared with Wt when these cells were treated either with leucine alone or in the presence of cysteine, suggesting a defect in leucine metabolism in ncl1Δ strain (Fig 5B). Leucine is metabolized to α-ketoisocaproate (KIC) through branched chain aminotransferase (BAT) (38). The expression of BAT1 was about twofold lower in ncl1Δ strain than Wt in the presence of cysteine (Fig 5C), and the levels of KIC was markedly reduced in cysteine-treated ncl1Δ strain even in the presence of leucine (Fig 5D). This confirms that the conversion of leucine to KIC is inhibited in ncl1Δ strain in the presence of cysteine, which may explain the inability of leucine to abrogate cysteine-induced growth defect in this strain. To confirm the role of KIC during cysteine treatment, we analyzed the effect of KIC on growth in Wt and ncl1Δ strains. Interestingly, although leucine could not rescue cysteine toxicity in ncl1Δ strain, KIC could rescue the toxicity in both

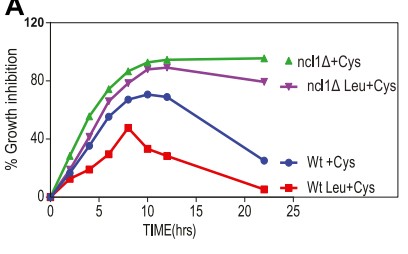

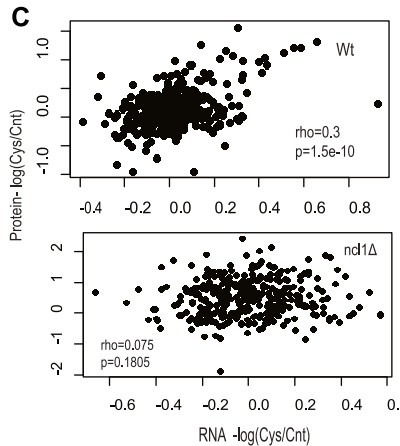

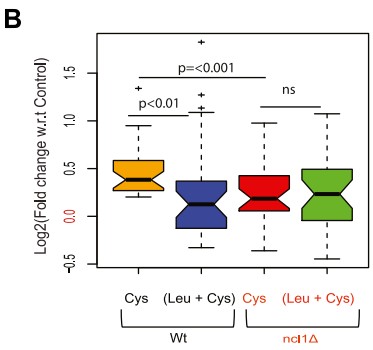

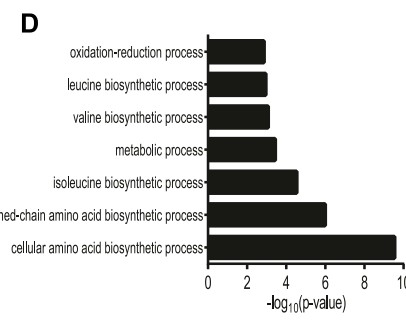

**Figure 4. Leucine reprograms proteome through NCl1 to recover cysteine-mediated toxicity.**
**(A)** Percentage growth inhibition of wild-type (Wt) and ncl1Δ strain, in the presence of cysteine and combination of leucine and cysteine. **(B)** Box plot represents the relative expression of proteins during co-treatment of leucine and cysteine in Wt and ncl1Δ, for proteins that were up-regulated by cysteine in Wt. Each condition represents the log2-fold change of different proteins (n = 32), which is an average of three biological replicates, and statistical significance was analyzed using paired nonparametric method. **(C)** Spearman rank correlation between relative expression of mRNA (obtained from RNA-seq) and protein levels (from iTRAQ-based relative proteomics) in Wt (upper panel) and ncl1Δ (lower panel). **(D)** The bar graph represents the biological pathways enriched from proteins whose expression was reverted during co-treatment of leucine and cysteine in Wt, but not to the similar extent in Δncl1. Classification was done using Database for Annotation, Visualization & Integrated Discovery (DAVID), and pathways with *P*-value ≤ 0.05 are plotted in the figure.

ncl1Δ and Wt strains (Fig 5E). This further proves that conversion of leucine to KIC is a necessary step for survival in the presence of cysteine. Interestingly, KIC was able to rescue the growth inhibition of other toxic amino acids similar to leucine (Fig 5F).

Both Keto-isocaproate (KIC) and leucine have been shown to activate TOR (39), and Bat1 deletion leads to compromised TORC1 activity (39). Bat1 exhibits leucine-dependent interactions with the tri carboxylic acid (TCA) enzyme aconitase, and deletion of Bat1 leads to TCA cycle block (39). Thus, we measured the levels of TCA intermediates in ncl1Δ strain during cysteine treatment and its co-treatment with leucine or KIC. We found that cysteine treatment resulted in accumulation of citrate, with decreased levels of other TCA intermediates (α-ketoglutarate, succinate, and oxaloacetate) (Fig 5G). However, KIC but not leucine could reverse the levels of most of these metabolites. Citrate accumulation could potentially decrease the levels of pyruvate (40), which prompted us to check for the levels of pyruvate during cysteine treatment. We found that cysteine treatment significantly lowered the levels of pyruvate in ncl1Δ strain (Fig 6A).

To understand the significance of pyruvate levels, we measured the growth of Wt and ncl1Δ cells during exogenous addition of pyruvate. In Wt cells, pyruvate (5 mM) was able to decrease the growth inhibition because of cysteine by about 35% (Fig 6B). Leucine at the same concentration was able to decrease the growth inhibition by about 45%. In ncl1Δ strain, where leucine could not abrogate the growth inhibition of cysteine, pyruvate could decrease the inhibition by more than 60%. Interestingly, we found that leucine and pyruvate had a synergistic effect and in combination could almost completely abrogate the cysteine-induced inhibition (Fig 6C), suggesting that leucine and pyruvate could have independent mechanisms for rescuing cysteine toxicity.

# Discussion

In this study, we show that excess of a single amino acid can lead to large-scale changes in cellular amino acid pool and rewire metabolism. As a paradigm, we show that a cysteine-induced change in metabolism is adaptive and is primarily routed through a translational regulator. We have used genetic, proteomic, transcriptomic, and targeted metabolomics approach to get systemic understanding of a cell exposed to metabolic stress (because of excess cysteine).

## Cysteine inhibits translation and rewires amino acid metabolism

Cell maintains homeostasis during cysteine accumulation by rewiring the amino acid metabolism and depends on known regulators of amino acid metabolism, Leu3 (41) and Stp2 (42), for survival. Cysteine increases the levels of lysine and decreases the levels of threonine, and as a consequence, the expression of enzymes required for their biosynthesis is altered in opposite directions. Lysine levels have been reported to be high in the yeast strains defective in translational machinery (43); thus, the levels of lysine may be a marker of translational defect. Cysteine decreases the levels of threonine, and genetic inhibition of threonine biosynthesis further increases the growth sensitivity toward cysteine. The involvement of threonine biosynthesis during high levels of cysteine was also reported in *Escherichia coli* more than three decades ago—where it was suggested that cysteine inhibits homoserine dehydrogenase (Hom6 in yeast) (44), acetohydroxyacid synthetase (Ilv6 in yeast) (45), and threonine deaminase (Ilv1 in

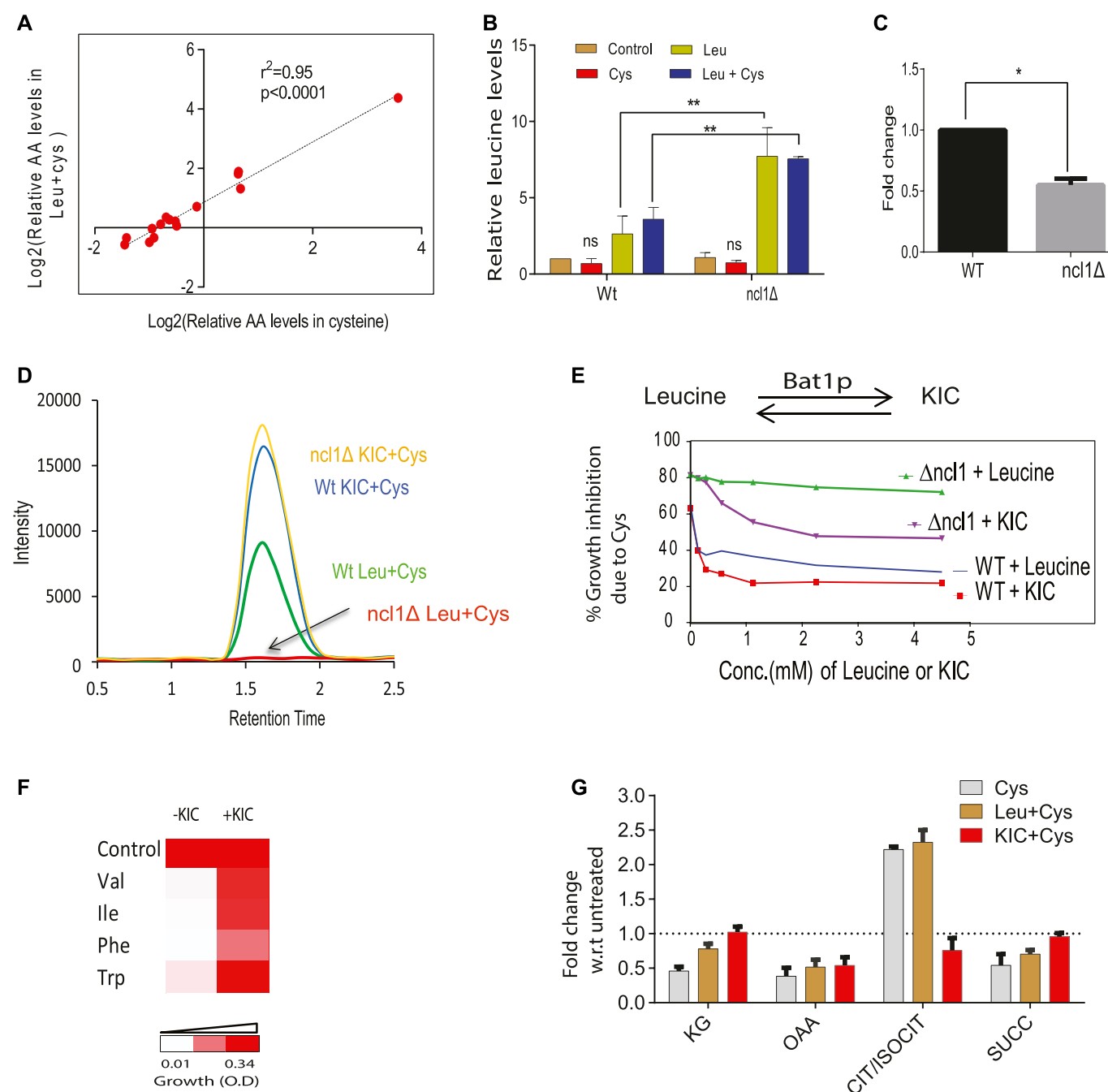

**Figure 5. Metabolism of leucine via NCL1 plays a vital role during cysteine-induced stress.**
**(A)** Relative amino acid (AA) levels in ncl1Δ strain during Cys treatment and combination of Leu and Cys treatment were measured and were analyzed by calculating the correlation between these two groups. **(B)** Bar graph represents the relative levels of leucine in Wt and ncl1Δ cells, during leucine and combination of leucine and cysteine, normalized with respect to levels of leucine in untreated Wt strain. Error bar represents SD (n = 2). Statistical significance was determined by unpaired *t* test. "**" indicates *P* < 0.01. **(C)** Relative expression of BAT1 in Wt and ncl1Δ during cysteine treatment, normalized with respect to its expression in cysteine-treated Wt. Error bar represents SD (n = 3). Statistical significance was determined by paired *t* test "*" indicates *P* < 0.05. **(D)** KIC levels were measured using MRM-based LC–MS technique. The figure represents the overlapped peak intensity of a transition of KIC (121/101 m/z), measured from Wt and ncl1Δ cells during co-treatment of leucine and cysteine. KIC was also measured in both these cells during co-treatment of KIC and cysteine. **(E)** Cells were co-treated with cysteine (1 mM) and different concentrations of leucine or KIC, and then growth was measured after 12 h. The graph represents the growth inhibition of Wt and ncl1Δ strain because of cysteine in the basal media and during its co-treatment with varying concentration of KIC and leucine. **(F)** Growth of cells in the presence of toxic amino acids (Ile, Val, Phe, and Trp) and during their respective co-treatment with KIC. **(G)** TCA intermediates were measured by MRM-based LC–MS technique. The bar plot represents their relative levels during cysteine treatment and its combination with KIC or leucine in ncl1Δ strain. Error bar represents SD (n = 3).

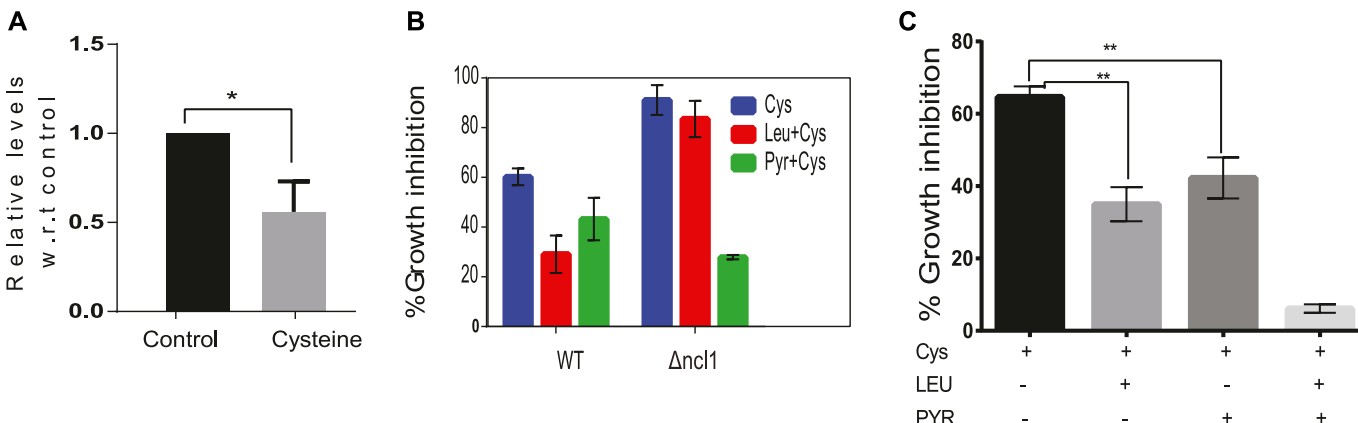

**Figure 6. Pyruvate and leucine co-treatment completely rescues the toxicity of cysteine.**
**(A)** Relative levels of pyruvate in ncl1Δ cells treated with cysteine, with respect to corresponding untreated control. Error bar represents SD (n = 3). Statistical significance was determined by paired *t* test. "*" indicates *P* < 0.05. **(B)** Effect of pyruvate and leucine on the growth inhibition induced by cysteine in Wt and ncl1Δ cells. Error bar represents SD (n = 3). **(C)** Growth inhibition of cysteine during co-treatment of leucine and pyruvate simultaneously. Error bar represents SD (n = 3). Statistical significance was determined by paired *t* test. "**" indicates *P* < 0.01.

yeast) (46). Furthermore, cysteine increases the levels of arginine and the enzymes involved in its biosynthesis, and this observation is in agreement with a recent report, suggesting that high levels of cysteine increase arginine biosynthesis for polyamine synthesis (47).

This study highlights the role of amino acid metabolism and metabolic genotype of strain in governing the cellular growth in the presence of high levels of cysteine. We have also shown that cell may need high levels of leucine to survive during stress induced by excess amino acids. Leucine not only reverses the expression of proteins induced by cysteine treatment but also maintains balanced amino acid levels. Consistent with the reports that leucine assists translation (48, 49), we also found that leucine rescues cysteine-induced translational defect. To understand if there is a significant alteration in the expression of proteins involved in amino acid metabolism, we performed a detailed proteomic study at different time points (15, 45, 120, and 360 min) of cysteine treatment. We did not find significant alteration in the expression of proteins involved in amino acid metabolism at 15 or 45 min, although they showed a higher trend. However, at 120 min, most of the amino acid metabolism proteins were up-regulated (data not shown) as was in the case of 6 h (Fig 2A). Interestingly, ribosomal proteins showed up-regulation even from 15 min (data not shown), suggesting that cysteine-induced protein translation inhibition may be the initial event, followed by imbalance in the levels of amino acids.

### NCL1-mediated translation rewires metabolism in the presence of high cysteine

It was found that NCL1, a methyltransferase, is required for survival in the presence of high levels of cysteine. Appropriate methylation marks have a vital role in the normal development, and loss of NCL1 homolog NSUN2 represses global protein synthesis and leads to neurodevelopmental defects in mouse and humans (50, 51, 52). Here, we have shown that in the presence of high levels of cysteine,

deletion of NCl1 abrogates the correlation between the mRNA and protein levels and depletes the levels of proteins involved in translational machinery. The exact role of NCL1 in translation modulation is unknown; however, it has been shown to have an important role in the TTG codon–mediated selective translation of proteins during oxidative stress (37). Most importantly, our results clearly indicate that NCl1 deletion inhibits the conversion of leucine to KIC, and supplementation of KIC but not leucine could rescue the cysteine-induced growth defect in this strain. Both leucine and KIC, via BAT1, could activate TORC1 independent of EGO1 complex. It is known that deletion of BAT1 leads to TCA cycle block (39). In this study, we have shown that expression of BAT1 depends on NCL1, and in ncl1Δ strain, cysteine leads to TCA cycle block, which can be attenuated by KIC and not by leucine.

### Relation between cysteine and pyruvate

Our results indicate that cysteine treatment increases the levels of citrate, a known inhibitor of pyruvate kinase (40). It has also been reported that cysteine directly inhibits pyruvate kinase and decreases the levels of pyruvate (53). Accumulation of phosphoenolpyruvate, the immediate precursor of pyruvate in cysteine-treated cells (data not shown), also points toward lower conversion of phosphoenolpyruvate to pyruvate, leading to low levels of pyruvate observed in our study. Low pyruvate levels could account for partial growth inhibitory effect of cysteine because addition of pyruvate and leucine together could completely abrogate the growth defect. Based on this and earlier studies (39), we speculate that addition of excess cysteine lowers the levels of pyruvate thereby affecting TCA cycle. Furthermore, the strain used in this study is auxotrophic for leucine, which could potentially inhibit the interaction of Bat1 with aconitase (because the interaction is leucine dependent), resulting in compromised TCA cycle. Interestingly, in higher eukaryotes, leucine is metabolized to acetyl-coA and enters into the TCA cycle (54). However, in yeast, the existence of this pathway is still not

well established. Thus, the exact manner in which cysteine, leucine, and pyruvate could interact with the TCA cycle intermediates thereby controlling TCA cycle is still a matter of conjecture and needs to be studied in detail. Increase in the concentration of cysteine leads to obesity and decreased tolerance (55); its prolonged treatment lowers the levels of pyruvate and inhibits glucose-induced ATP production in pancreatic cells (53, 56). Also, in a case–control study in the Indian population, we have previously shown that high levels of cysteine is associated with cardiovascular disease (32), and interestingly, total proteins in coronary artery disease patients was shown to be significantly lower as compared with control groups (57). A diet high in leucine content has been proposed to be helpful in attenuating heart diseases (58). Extrapolating these observations, we speculate that the association of elevated cysteine levels with metabolic disease may be linked through energy metabolism and inhibition of protein translation, which could be restored by leucine supplementation.

# Materials and Methods

## Materials

Yeast media constituents, including yeast extract, peptone, and dextrose, were purchased from Himedia (India), and the amino acids were purchased from Sigma-Aldrich. Iodoacetamide, DTT, formic acid, and ammonium formate were bought from Sigma-Aldrich, and sequencing grade trypsin was procured from Promega. Other MS-based reagents and columns were purchased from Sciex, as mentioned previously (59, 60).

## Yeast strains, media, and growth conditions

The wild-type *S. cerevisiae* strains BY4741 (MATa his3Δ1 leu2Δ0 met15Δ0 ura3Δ0) and S288C used in this study were procured from American Type Culture Collection. The yeast deletion library in the background of BY4741 was obtained from Invitrogen. In S288C background, leu2Δ strain was generated using homologous recombination. Preinoculation of all the strains was performed in YPD media composed of yeast extract (1%), peptone (2%), and dextrose (2%). The experiments were performed in SC media containing glucose (2%), yeast nitrogen base (0.17%), ammonium chloride (0.5%), adenine (40 μg/ml) uracil (20 μg/ml), and amino acids (24). The amino acids mixture of the SC media composed of valine (150 μg/ml), threonine (200 μg/ml), arginine–HCl (20 μg/ml), phenylalanine (50 μg/ml), tyrosine (30 μg/ml), leucine (60 μg/ml), aspartic acid (100 μg/ml), lysine (30 μg/ml), histidine (20 μg/ml), glutamic acid—monosodium salt (100 μg/ml), and tryptophan (40 μg/ml) (24, 59). In the growth screen (Figs 1, 2, and 3), the amino acids were added in the concentration of 10 times higher than mentioned in the SC media. However, for the amino acids that were not present in SC media, an arbitrary concentration of 4 mM (close to the average concentration of all the amino acids, 4 mM) for alanine, glutamine, isoleucine, asparagine, glycine, serine, and proline, and 1 mM for cysteine was used.

## Growth assay

Yeast cells were preinoculated in YPD media overnight in an incubator shaker (Thermo Fisher Scientific) at 30°C and 200 rpm. The saturated culture was then washed three times with sterile water and reinoculated at 0.1 OD in SC media. To study the effect of exogenously added amino acids, cells were treated with excess amino acids and were grown at 30°C in an incubator shaker at 200 rpm and then growth was measured after 12 h. Growth kinetics with cysteine treatment was monitored by withdrawing aliquots at different time points and measuring the turbidity (at 600 nm) using a spectrophotometer (Eppendorf Biophotometer plus).

## Drop dilution assay

Drop dilution assay was performed to investigate the viability of cells because of treatment of excessive amino acids. Overnight saturated yeast culture in YPD was washed three times with sterile water and reinoculated at 0.1 OD in SC media and treated with different amino acids for 12 h. The cells were then harvested by centrifugation and washed twice by sterile water and diluted to $3 \times 10^7$ colony-forming units. Subsequent 10-fold dilutions were made and 4 μl were spotted on SD agar plate.

## Cell size determination by flow cytometry

Yeast cells treated with different amino acids were harvested and washed twice by sterile water and suspended in 0.5 ml of sterile water and acquired in BD Accuri C6 flow cytometer (BD Biosciences). Forward scattering were measured for 30,000 events per sample after gating the control cells.

## Yeast knock out

All yeast deletion strains in the background of BY4741 were obtained from Yeast Knock Out Library. The deletion of LEU2 in S288C background was carried with NAT cassette having overhangs for homologous recombination (61) using 5′ primer and 3′ primer. The NAT cassette was amplified from pYMN23 plasmid, using forward primer (AAATGGGGTACCGGTAGTGTTAGACCTGAACAAGGTTTACAGCTT-CGTACGCTGCAGGTC) and reverse primer (TTAAGCAAGGATTTTCTTA-ACTTCTTCGGCGACAGCATCAGCTTCTAA TCCGTACTAGAG). The cassette was transformed in S288C strain, using LiAc yeast transformation as described below and selected for NAT resistance.

## Yeast transformation

Yeast cells were grown to 0.6 OD and then harvested by centrifugation at 2,500g for 5 min at 30°C. Later, cells were resuspended in lithium acetate (LiAc) buffer and incubated in shaker incubator for 30 min and then centrifuged at 2,500g for 5 min at 300°C to pellet down the cells. Cells were then resuspended in 100 μl of LiAc buffer, and the template DNA (2 μg) and single-stranded carrier DNA (5 μg) were added. Then, 50% PEG with LiAc buffer was added and vortexed briefly and incubated at 30°C for 90 min. Then heat shock was given for 10 min at 42°C. The solution was then centrifuged at 9,000 g for 1 min and the supernatant was

discarded. The pellet was washed with autoclaved sterile water. The pellet was then resuspended in YPD for 1 h at 30°C with gentle shaking at 120 rpm and 100–150 $\mu$l of total sample was plated on sterile YPD-Nat plates.

### Polysome profiling

Yeast cells were grown in the presence and absence of 1 mM cysteine for 6 h, and then 50 $\mu$g/ml of cycloheximide was added to the cultured cells and kept in ice for 5 min to stall the ribosomes. Cells were then processed as previously (60). Briefly, cells were harvested at 2,500$g$ for 5 min at 4°C, and then lysis was performed with bead beater in lysis buffer (50 mM Tris, pH 7.5, 150 mM NaCl, 30 mM MgCl$_2$, and 50 $\mu$g/ml CHX). RNA content of the samples was normalized by measuring the absorbance at 260 nm. Equal amount of RNA (5 OD at 260 nm) was then loaded for each of the sample to the 7–47% continuous sucrose gradient (50 mM Tris acetate, pH 7.5, 250 mM sodium acetate, 5 mM MgCl$_2$, 1 mM DTT, and 7–47% sucrose) and centrifuged in Beckman SW32Ti rotor at 120,000 $g$ for 4 h at 4°C. The samples were then fractionated using ISCO gradient fractionator with detector sensitivity set to 0.5. This fractionator was equipped with a UV absorbance monitor, and thus, optical density of the gradients at 254 nm was measured in real time to obtain the profile.

### 35S-Methionine labeling

De novo protein translation was measured by the rate of incorporation of 35S-methionine in the nascent polypeptides. Yeast cells are grown in the SC media for 0.6–0.8 OD and then washed with sterile water. Cells were resuspended in SC media without methionine for 10 min, and then finally 35Smethionine and cold methionine were added. Then cells were distributed in three groups: one group was kept as control, and in other two groups, 1 and 2 mM cysteine was added, respectively. Then cells were incubated at optimum temperature, and 1 ml aliquots were withdrawn at different time points: 15, 45, 120, and 360 min.

Cells were lysed using alkaline lysis method (62) and isolated protein was estimated using BCA (bicinchoninic acid) method. Equal protein was then loaded on SDS–PAGE and stained with Coomassie blue staining. Incorporation of 35S-methionine in the isolated proteins was then analyzed by scanning the protein gel using ImageSafe scanner.

### Protein isolation for MS-based proteomics

Cells with and without cysteine treatment were harvested and washed three times with sterile water. As mentioned previously (59, 60), lysis was performed by bead beating method using lysis buffer containing 10 mM Tris–HCl (pH 8), 140 mM NaCl, 1.5 mM MgCl$_2$, 0.5% NP40, and protease inhibitor cocktail. Isolated protein was then buffer exchanged with 0.5 M triethyl ammonium bicarbonate (pH 8.5) using a 3-kD cutoff centrifugal filter (Amicon, Millipore). Protein was then estimated using Bradford reagent (Sigma-Aldrich).

### Trypsin digestion and Isobaric tag for relative and absolute quantitation (iTRAQ) labeling

Buffer exchanged proteins from different groups were digested using trypsin as mentioned previously (59, 60, 63). Briefly, 60 $\mu$g of protein from each sample was reduced with 25 mM DTT for 30 min at 56°C and the cysteines were blocked by 55 mM IAA at room temperature for 15–20 min. This was followed by digestion with modified trypsin added in the ratio of 1:10 ratio (trypsin to protein) and incubated at 37°C for 16 h. We used 4-plex and 8-plex iTRAQ labeling strategies. In the 4-plex experiment, the proteins digests from cells treated with and without cysteine for 12 h were used. However, in the 8-plex experiment, protein from Wt and ncl1Δ strains treated with cysteine and combination of leucine and cysteine for 6 h was used. With biological triplicates, both the labeling procedures were performed using the manufacturer's protocol (AB Sciex) (59, 60, 63).

### 2D nano LC–MS/MS iTRAQ analysis

iTRAQ labeled peptides were first fractionated by cation exchange (SCX) chromatography with SCX Cartridge (5 $\mu$m, 300 Å bead from AB Sciex) using a step gradient of increasing concentration of ammonium formate (30, 50, 70, 100, 125, 150, and 250 mM ammonium formate, 30% vol/vol ACN, and 0.1% formic acid; pH = 2.9) (59, 60, 63). The iTRAQ labeled fractions of 4-plex and 8-plex obtained after SCX chromatography were analyzed on quadrupole-TOF hybrid mass (Triple TOF 5600 & 6600; Sciex) spectrometer coupled to an Eksigent NanoLC-Ultra 2D plus system.

For each fraction, 10 $\mu$l of sample was loaded onto a trap column (200 $\mu$m × 0.5 mm) and desalted at flow rate 2 $\mu$l/min for 45 min. Then peptides were separated using a nano-C18 column (75 $\mu$m × 15 cm) using a gradient method with buffer A (99.9% LC-MS water + 0.1% formic acid) and buffer B (99.9% acetonitrile + 0.1% formic acid) as described previously (59, 60, 63). Data were acquired in an information-dependent acquisition (64) mode with MS settings as follows: nebulizing gas of 25, a curtain gas of 25, an ion spray voltage of 2,400 V, and heater interface temperature of 130°C. TOFMS scan was performed in the mass range of 400–1,600 m/z with accumulation time of 250 ms, whereas the MS/MS product ion scan was performed with mass range of 100–1,800 m/z for 70 ms with a total cycle time of ~2.05 s. Parent ions with more than 150 cps abundance and with a charge state of +2 to + 5 were selected for MS/MS fragmentation. After an MS/MS fragmentation of an ion, its mass and isotopes were excluded for fragmentation for 3 s. MS/MS spectra were acquired using high sensitivity mode with "adjust collision energy when using iTRAQ reagent" settings (59, 60, 63).

### Database searching and analysis for proteomics experiment

The raw files in the format of ".wiff" containing the spectra of MS and MS/MS were submitted to Protein Pilot v4.0 software (AB Sciex) for protein identification and relative quantification. Paragon Algorithm was used in a "Thorough ID" search mode against the Uniprot *S. cerevisiae* reference dataset (6,643 protein sequences). The parameter for identification search includes IAA as cysteine blocking agent, 4-plex or 8-plex N-terminal iTRAQ labeling, and trypsin

digestion with two missed cleavages. Global protein level 1% false discovery rate was considered for protein identification (59, 60, 63).

In 4-plex experiment, we identified 1,041 proteins in all the three biological replicates with a criteria of 1% global false discovery rate and more than or equal to two unique peptides (59). Proteins were considered to be differentially expressed if the fold change with respect to control was ≥2 or ≤0.5 in all the three replicates. With the similar criteria of identification, 543 proteins were identified in all the three biological replicates of 8-plex experiment. Based on fact that fold change of protein expression is generally underestimated in 8-plex iTRAQ experiment (64, 65), and the narrow range of fold change in the present data set, a fold change cutoff of ±1.15 was considered (66, 67). Thus, proteins were considered to be differentially expressed if the average fold change was ≤0.85 or ≥1.15, with a similar trend in at least two replicates.

### Amino acid profiling

Intercellular amino acids were extracted using boiling water lysis method (68, 69). Briefly, cells (5 OD) were washed three times with MilliQ water, and then reconstituted in 500 $\mu$l sterile water, and boiled for 10 min. Cell debris was removed by centrifugation for 5 min at 3,500 $g$. Amino acids were then measured by automated precolumn o-Phthaldialdehyde (OPA) derivatization using HPLC (1290 Infinity LC system; Agilent Technologies Pvt. Ltd). The mobile phases consisted of 10 mM $Na_2HPO_4$ and 10 mM $Na_2B_4O_7$ (buffer A) and acetonitrile-methanol-$H_2O$ in the ratio of 45:45:10 by volume (buffer B). In the automated preconditional method, 1 $\mu$l from sample was mixed with 2.5 $\mu$l of borate buffer (Agilent P/N 5061-3339), followed by addition of 0.5 $\mu$l of OPA dye (Agilent P/N 5061-3335). Finally, 32 $\mu$l of diluent (1 ml buffer A + 15 $\mu$l phosphoric acid) was added and 6 $\mu$l of mixture was loaded, and separated using Poroshell HPH-C18 column (2.7 $\mu$m, 3.0 × 100 mm). Amino acids were eluted from the column at a rate of 0.9 ml/min for 14.5 min, using a linear gradient of 2–57% of buffer B for 13.5 min. The eluted amino acids were detected by the fluorescence detector (Ex = 340 nm, Em = 450 nm).

### Genetic screen for growth sensitivity

A commercially available S. cerevisiae library of ~4,500 nonessential genes was used. All the deletion strains in this library contain a specific gene deleted, with KanMX resistance cassette. Primary culture was obtained by growing these strains overnight in 200 $\mu$l of YPD media containing G418, in 96-well format. Then, 5 $\mu$l of this culture was transformed to another 96-deep-well plate containing 400 $\mu$l of SC media, with and without 1 mM cysteine. The strains were then grown at 30°C in a shaking incubator (Thermo Fisher Scientific), with 200 rpm, for 12 h, and then, cell density was measured using multimode reader.

### RNA isolation

RNA was isolated from both Wt and ncl1Δ, treated with and without 1 mM cysteine for 6 h. Acid–phenol–based method was used to isolate total RNA and then purified using RNAeasy mini kit (QIAGEN) after a DNAse treatment (TURBO DNase; Ambion) following the manufacturer instruction (70). The RNA was examined on ethidium bromide–stained agarose gel to confirm its integrity.

### Library generation and mapping of sequencing reads

Library was prepared with 700 ng RNA of each sample using Truseq RNA sample prep kit v2 (stranded mRNA LT kit) according to the manufacturer's instructions (Illumina Inc.) (71 Preprint). AMPure XP beads (Agencourt) were used to purify adaptor-ligated fragments and then amplified (12–14 cycles), purified, and finally measured using Qubit instrument (Invitrogen). Then, using BioAnalyzer DNA1000 LabChip (Agilent Technologies), the average fragment size of the libraries was determined. And finally, the diluted libraries were multiplexed and loaded on HiSeq Flow Cell v3 (Illumina Inc.). Sequencing runs were performed on a HiSeq 2000 Illumina platform using TruSeq SBS Kit v3 (Illumina Inc.) for cluster generation as per manufacturer's protocol.

Sequencing reads with Phred quality score equal or greater than 30 were taken for analysis. Read sequences were trimmed using Trimmomatic (v0.43), and then aligned to the transcriptome of S. cerevisiae strain S288C as available from ENSEMBL using Kallisto (v0.36) software. Gene expression levels were then estimated as transcripts per million values.

### Targeted metabolomics

Metabolites for MS-based targeted metabolomics were extracted using cold methanol method. Cells (50 OD) from different treatment groups were washed three times with sterile water, and then quenched with chilled ethanol (kept at –80°C), and followed by bead beating using acid-washed glass beads. The suspension was then transferred to a fresh tube and centrifuged at 15,000 $g$ at 4°C for 10 min. Supernatant was then vacuum dried and then reconstituted in 50 $\mu$l of 50% methanol. The reconstituted mixture was centrifuged at 15,000 $g$ for 10 min, and 5 $\mu$l was injected for LC–MS/MS analysis.

The data were acquired using a Sciex Exion LCTM analytical UHPLC system coupled with a triple quadrupole hybrid ion trap mass spectrometer (QTrap 6500; Sciex) in a negative mode. Samples were loaded onto an Acquity UPLC BEH HILIC (1.7 $\mu$m, 2.1 × 100 mm) column, with a flow rate of 0.3 ml/min. The mobile phases consisted of 10 mM ammonium acetate and 0.1% formic acid (buffer A) and 95% acetonitrile with 5 mM ammonium acetate and 0.2% formic acid (buffer B). The linear mobile phase was applied from 95% to 20% of buffer A. The gradient program was used as follows: 95% buffer B for 1.5 min, 80%–50% buffer B in next 0.5 min, followed by 50% buffer B for next 2 min, and then decreased to 20% buffer B in next 50 s, 20% buffer B for next 2 min, and finally again 95% buffer B for next 4 min. Data were acquired using three biological triplicates with three technical replicate for each run. A list of measured metabolites and their optimized parameters are given in the table (Table S8). Relative quantification was performed using MultiQuantTM software v.2.1 (Sciex).

Pyruvate measurement was done using syringe pump–based direct infusion method. Lyophilized samples were reconstituted in 250 $\mu$l of 50% methanol. Harvard syringe pump coupled with 6500

QTRAP in negative mode was used. Infusion parameters were set as follows: ion spray voltage: −4,500 V, CUR gas: 35, GS1: 20, GS2: 10, CAD: high, DP: −60 V, CE: −38 V, and flow rate: 7 $\mu$l/min. Multiple reaction monitoring (MRM) transition used for pyruvate was 87/43 and peak intensities obtained was used for relative quantification.

## Data availability

Transcriptomic data generated in the article has been submitted to SRA database under the accession number PRJNA514239 The mass spectrometry proteomics data have been deposited to the ProteomeXchange Consortium (http://proteomecentral.proteomexchange.org) via the PRIDE partner repository with the dataset identifier PXD014859 (for 4 plex) and PXD014880 (for 8 plex)

## Supplementary Information

## Acknowledgements

We acknowledge the financial support from the Council of Scientific and Industrial Research, India. The study was funded under the project titled "CARDIOMED: Centre for Cardiovascular and Metabolic Disease Research (BSC0122)". R Chakraborty acknowledges the Junior Research Fellowship from UGC. We thank Dr. Shuvadeep Maity and Dr. Tryambak Basak for their contribution in standardizing yeast and proteomics experiments, respectively. We are grateful to Asmita Ghosh and Sarada Das for transcriptomic experiments. We thank Dr. Vignesh Kumar for his help in standardizing polysome profiling experiment. We are grateful to Meghali Aich for flow cytometry experiment. We also thank Dr. Munia Ganguly and Dr. Arjun Ray for proof reading the manuscript. We are also thankful to Zeeshan Hamid and all master students for being a valuable part of this project.

### Author Contributions

A Bhat: conceptualization, data curation, formal analysis, visualization, methodology, and writing—original draft, review, and editing.
R Chakraborty: data curation, formal analysis, methodology, and writing—original draft.
K Adlakha: data curation, formal analysis, and methodology.
G Agam: data curation, formal analysis, and methodology.
K Chakraborty: conceptualization, data curation, supervision, investigation, methodology, and writing—original draft, review, and editing.
S Sengupta: conceptualization, data curation, supervision, funding acquisition, investigation, visualization, methodology, and writing—original draft, review, and editing.

### Conflict of Interest Statement

The authors declare that they have no conflict of interest.

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
