## [Reviewer comments · Life Science Alliance]

Life Science Alliance

Ncl1 mediated metabolic rewiring is critical during metabolic stress

Ajay Bhat, Rahul Chakraborty, Khushboo Adlakha, Ganesh Agam, kausik Chakraborty, and Shantanu Sengupta

DOI: <https://doi.org/10.26508/lsa.201900360>

Corresponding author(s): Shantanu Sengupta, Institute of Genomics and Integrative Biology and kausik Chakraborty, CSIR-Institute of Genomics and Integrative Biology

Review Timeline:

Submission Date:	2019-02-25
Editorial Decision:	2019-03-18
Revision Received:	2019-07-05
Editorial Decision:	2019-07-22
Revision Received:	2019-08-02
Accepted:	2019-08-05

Scientific Editor: Andrea Leibfried

Transaction Report:

March 18, 2019

Re: Life Science Alliance manuscript #LSA-2019-00360-T

Shantanu Sengupta
Institute of Genomics and Integrative Biology
Proteomics and Structural Biology Unit
Mall Road
Delhi, Delhi 110007
India

Dear Dr. Sengupta,

Thank you for submitting your manuscript entitled "Ncl1 mediated metabolic rewiring is critical during metabolic stress" to Life Science Alliance. The manuscript was assessed by expert reviewers, whose comments are appended to this letter.

As you will see, the reviewers appreciate your work, but they think that more insight into the feedback reactions is needed as well as additional controls and clarifications. We would thus like to invite you to submit a revised version of your work, addressing the individual concerns raised by the reviewers. Importantly, both reviewers also note that your manuscript needs language editing, and this concern should get thoroughly addressed.

Thank you for this interesting contribution to Life Science Alliance. We are looking forward to receiving your revised manuscript.

Sincerely,

Andrea Leibfried, PhD
Executive Editor
Life Science Alliance
Meyershofstr. 1
69117 Heidelberg, Germany
t +49 6221 8891 502
e a.leibfried@life-science-alliance.org
www.life-science-alliance.org

B. MANUSCRIPT ORGANIZATION AND FORMATTING:

Reviewer #1 (Comments to the Authors (Required)):

All living cells respond to external stimuli such as nutrients and maintain homeostasis in response to the depletion or excess of factors, which are indispensable for the cells' viability. It is rather well known how cells sense the reduction in the amino acid level and which signaling pathway is

involved in this process. mTORC1 and GCN2 are the best-known proteins (complex) to regulate cellular anabolic/catabolic processes for the maintenance of cellular homeostasis in response to the deprivation of amino acids. On the other hand, it is poorly understood how cells maintain their homeostasis under an excessive nutrient condition, while high amounts of many amino acids are known as toxic and associated with disease. In this manuscript, Bhat et al. addressed an interesting and important question as to how cells respond to the excessive nutrient level and protect themselves from the cytotoxicity. The authors suggested that in yeast, excessive cysteine, which suppresses cell growth and protein synthesis, but co-treatment with leucine alleviate growth inhibition via leucine biosynthesis pathway. Using a proper model and genetic, proteomic, and metabolic approaches, the authors investigated systemically what is altered by excessive cysteine supplementation and what is critical for the recovery from growth inhibition at the metabolic level. They identified that NCL1, m5C-methyltransferase, is responsible for the recovery from cysteine-mediated cell growth inhibition by rewiring cellular translation and metabolism. The authors' hypothesis is well supported by the results and their recovery experiments appear to work well. However, there is a need for more explanations of the feedback reactions in TCA cycle and TCA intermediates derived from cysteine and leucine. In addition, the concerns below should be addressed.

Major concerns:

1. English must be improved.
2. The authors suggest based on Fig. 1 that intracellular leucine concentration modulated by LEU2 is important for reducing cysteine-mediated cytotoxicity. Can the authors see a similar effect to that in Fig. 1A when they treat prototrophic yeast strain with excessive cysteine and restrained leucine? In Fig. 1B, does the pre-treatment of cysteine or leucine differ from the result of co-treatment of both amino acids? In addition, glutamate seems less potent than leucine in the recovery from cysteine-mediated growth inhibition, but seems to have a tryptophan- and leucine-specific recovery effect. Can the authors explain the effect of glutamate by comparing to that of leucine and show the involved metabolic pathway in Fig 2A?
3. Fig. 1, the authors focused on cell growth/proliferation in the presence of excessive amino acids, but how about cell size and cell death in the same treatment?
4. Figs. 2B and 2C, which time point did the authors use (early - 6 h or later - 12 h)? Did leucine supplementation up-regulate proteins such as HOM6, LYS4, ARG3 in later time points? In Fig. 2D, it seems the effect of excessive cysteine treatment on protein synthesis is rapid (significant difference within 35 min). Could the authors see any significant alteration in the expression of proteins involved in amino acid metabolism during the same period?
5. Fig 4A, the growth inhibition of wildtype strain is decreasing even in excessive cysteine conditions and seems to be normal after 30 hours. How can the authors explain this recovery?
6. Fig 4C, it is hard to detect that either transcription or translation is significantly affected in *ncl1Δ*. Can the authors fill the dots with the same color for the same gene in both graphs? In addition, the authors should show mRNA and protein levels for both genotypes in Cys+Leu/Cys. Is the decoupling between transcription and translation resulting from the known NCL1 function?
7. Fig 5B, can the authors show relative leucine levels for both cells under excessive cysteine only condition?

Minor concerns:

1. Matrices in Fig 1B and 1E can lead to misunderstanding as if blue is negative, and red is positive. Since the growth percentage is between 0 and 100 (+α), the color of the matrix should be changed to a linear gradient with a single color.
2. In Fig 4B, the color of the 1st box and 4th box is the same. In Fig 4D, the shape of the bar for 'branched-chain amino acid biosynthetic process' is different from others.

3. In Fig 5E, what is Batp above the arrows?
4. In Fig 5G, what does 'fold change' mean in y-axis?

Reviewer #2 (Comments to the Authors (Required)):

Bhat and co-authors present an analysis of the budding yeast metabolism in response to elevated cysteine levels in the medium. They show that leucine can partly abrogate the toxic effect of cysteine and they show that this rescue depends on Ncl1. Further, they show that the leucine metabolite KIC can rescue cysteine toxicity and that pyruvate downregulation in response to elevated cysteine levels is also involved in the toxicity.

This study is somewhat restricted because it focuses on the toxicity of a single amino acid (cysteine). However, it is scientifically sound and it is characterized by the application of a wide range of omics methods for the characterization of the cellular response to cysteine. The study presents data about a whole-genome screen for interactors mediating the leucine rescue effect, further proteomics data, transcriptomics data, metabolomics data, polysome profiling and various other assays. This is a pretty extensive study.

I only have one major comment: the language has to be substantially improved. There are many grammatical errors in the manuscript, which definitely has to be revised by a native speaker. Further, some descriptions and figure labels are unclear (see below). It is puzzling that one of the authors' name was misspelled in the submission system. One would hope that the rest of the analysis was performed with greater care.

1. Already at the beginning of the results section (page 3) it should be made clear which base-medium was used and how much higher than normal the AA concentrations were. When concentrations of AAs are given: do they refer to the amount added to the medium? Or do they refer to the total amount, i.e. what's already in the medium plus the added amount? Be more explicit and clearer.
2. Figure 1B: the color scale is confusing. When two colors are used they usually represent a reduction compared to normal and an increase compared to normal. Here, grey is not neutral, but some middle range growth, blue is low and red fast growth. That is counterintuitive.
3. Page 4, proteomics experiments: which strain was used? Which cysteine concentration? How does the concentration compare to normal (how much greater than normal)?
4. Page 5, genomic screen: provide more details on the screen in the results section. More is given in the methods part, but you should also give more details here. How many genes were tested? What was the readout for the assay? I.e. what exactly was tested?
5. Figure 3: How many strains (genes) were selected in each phase? Were the p-value corrected for multiple testing?
6. Still Figure 3A: The color red is used for two different things in the same figure, that is very confusing. In the middle panel it indicates resistant strains, in the bottom panel sensitive strains. Further, the labels are too small.

7. Figure 4B and Suppl. Fig. 3B: Which proteins are shown here? I assume only proteins showing an effect to cysteine alone are shown. But this needs to be made explicit. Based on which statistical criteria were the proteins selected?
8. Page 7, end of results: Leucine effects and pyruvate effects at equal concentrations are compared. Can one directly compare the molar concentrations between these two substances?
9. Page 9: spell out 'CAD'.
10. Page 13, proteomics analysis: differential expression of proteins is defined based on fold-changes only, but no proper statistics is performed. I think the authors should at least log-transform the intensities and then perform a t-test. Likely this will not change the conclusions much, but it is the proper way to do this.

Reviewer 1

Comment: All living cells respond to external stimuli such as nutrients and maintain homeostasis in response to the depletion or excess of factors, which are indispensable for the cells' viability. It is rather well known how cells sense the reduction in the amino acid level and which signaling pathway is involved in this process. mTORC1 and GCN2 are the best-known proteins (complex) to regulate cellular anabolic/catabolic processes for the maintenance of cellular homeostasis in response to the deprivation of amino acids. On the other hand, it is poorly understood how cells maintain their homeostasis under an excessive nutrient condition, while high amounts of many amino acids are known as toxic and associated with disease. In this manuscript, Bhat et al. addressed an interesting and important question as to how cells respond to the excessive nutrient level and protect themselves from the cytotoxicity. The authors suggested that in yeast, excessive cysteine, which suppresses cell growth and protein synthesis, but co-treatment with leucine alleviate growth inhibition via leucine biosynthesis pathway. Using a proper model and genetic, proteomic, and metabolic approaches, the authors investigated systemically what is altered by excessive cysteine supplementation and what is critical for the recovery from growth inhibition at the metabolic level. They identified that NCL1, m5C-methyltransferase, is responsible for the recovery from cysteine-mediated cell growth inhibition by rewiring cellular translation and metabolism. The authors' hypothesis is well supported by the results and their recovery experiments appear to work well. However, there is a need for more explanations of the feedback reactions in TCA cycle and TCA intermediates derived from cysteine and leucine. In addition, the concerns below should be addressed.

Response: We thank the reviewer for critically evaluating the manuscript and finding that our hypothesis is well supported by the results. As suggested by the reviewer we have tried to explain the TCA cycle vis-à-vis cysteine and/or leucine supplementation in the discussion section (page 10).

Comment: English must be improved.

Response: We sincerely apologize for this and have now tried to improve English and hope that it is up to the expectation.

Comment: The authors suggest based on Fig. 1 that intracellular leucine concentration modulated by LEU2 is important for reducing cysteine-mediated cytotoxicity. Can the authors see a similar effect to that in Fig. 1A when they treat prototrophic yeast strain with excessive cysteine and restrained leucine?

Response:

We had shown that excess cysteine in LEU2Δ strain inhibited the growth of yeast by about 80% (Figure 1A). However, in the prototrophic strain, in presence of leucine in the media, the growth inhibition was

about 40% (Figure 1C). As suggested by the reviewer, we checked the effect of excessive cysteine in prototrophic strain in the absence of leucine in the media and found no significant change in presence or absence of leucine in the media. This clearly indicates that an optimum concentration of leucine is required to alleviate the toxicity of cysteine. This concentration of leucine can be achieved either by intracellular synthesis or by extracellular supplementation. This has now been mentioned in the results section (page 4) and supplementary figure (Fig S1C).

Comment: In Fig. 1B, does the pre-treatment of cysteine or leucine differ from the result of co-treatment of both amino acids?

Response: We thank the reviewer for bringing up an interesting point. To check if pre-treatment of cysteine or leucine had an effect similar to the co-treatment of these amino acids, we grew yeast cells (BY4741) in presence of either cysteine or leucine for 6 hours, washed the media and treated the cells with either leucine or cysteine, respectively. In both the conditions, leucine was able to rescue cysteine-induced toxicity (Fig. S1D). This has now been mentioned in the results section page 4 and shown in supplementary figure (Fig S1D)

Comment:

In addition, glutamate seems less potent than leucine in the recovery from cysteine-mediated growth inhibition, but seems to have a tryptophan- and leucine-specific recovery effect. Can the authors explain the effect of glutamate by comparing to that of leucine and show the involved metabolic pathway in Fig 2A?

Response

As mentioned by the reviewer, Glutamate was less potent than leucine in recovering cysteine-mediated growth defect. Therefore, the focus of the manuscript was to understand the mechanism of leucine mediated recovery of cysteine induced toxicity. However, to address the reviewer's query we performed a proteomics experiment after adding cysteine or cysteine+glutamate to yeast cells for 12 hours and compared the expression of proteins in the amino acid pathway with the effect of cysteine+leucine. We found that unlike leucine, glutamate could not rescue the altered protein expression induced by cysteine (Shown in the figure below). Since understanding the effect of glutamate was not in the scope of the study we have not included this in the revised manuscript. However, if the reviewer feels we can include this.

Effect of Cys, Leu and Glu on the expression of proteins involved in amino acid metabolism

Comment: Fig. 1, the authors focused on cell growth/proliferation in the presence of excessive amino acids, but how about cell size and cell death in the same treatment?

Response: We thank the reviewer for this comment. To address this question we have now measured cell size and cell death of wild type strain BY4741 in presence of seven different amino acids (which include both toxic and nontoxic amino acids). For cell size, we have measured the forward scattering by flow cytometer. Out of these seven amino acids, none of them (including toxic amino acids) altered the size of the cells.

For cell death experiment, we have performed a spotting assay in presence of all of the seven amino acids used in the cell size experiment. Interestingly, we found that Cysteine and Tryptophan have a “static” effect whereas phenyl alanine and isoleucine have a “cidal” effect. While the non-toxic amino acids showed normal growth.

We have now included this in the revised manuscript (page 3) and figure S1A and S1B.

Comment: Figs. 2B and 2C, which time point did the authors use (early - 6 h or later - 12 h)? Did leucine supplementation up-regulate proteins such as HOM6, LYS4, ARG3 in later time points?

Response: We apologize for this confusion. In the Fig 2B, we have used early time point (6 hrs). However, in Fig 2C we have used later time point (12 hrs). As we found that proteins involved in amino acid metabolism were altered by cysteine at early time point, we chose the earlier time point to investigate the effect of leucine. However, the levels of amino acid altered to an appreciable extent at 12 hours but not at the early time point. This led us to check the effect of leucine on the amino acid levels at the later time point. As desired by the reviewer we checked the expression of HOM6, LYS4 and ARG3 at 12 hours and found that the expression of HOM6, which was downregulated after cysteine treatment reverted back to control levels on addition of leucine. However, there was no change of protein expression of LYS4 and ARG3 even after addition of leucine.

Comment: In Fig. 2D, it seems the effect of excessive cysteine treatment on protein synthesis is rapid (significant difference within 35 min). Could the authors see any significant alteration in the expression of proteins involved in amino acid metabolism during the same period?

Response: *Note: The s35 experiment as mentioned in the methods was done for 15, 45, 120 and 360 minutes. However, in its corresponding fig (Fig 2D), 15 minutes was mistyped as 35 minutes. We apologize for this mistake and the confusion caused by it. We have corrected the mistake in the revised manuscript.*

We agree with the reviewer that effect of cysteine treatment on protein synthesis seems to be rapid. To understand if there is significant alteration in the expression of proteins involved in amino acid

metabolism we performed a detailed proteomics experiment at different time points (15, 45, 120 and 360 minutes) of cysteine treatment. We did not find significant alteration in the expression of proteins involved in amino acid metabolism at 15 or 45 minutes, although they showed a higher trend. However, at 120 mins most of the amino acid metabolism proteins were up regulated as was in the case of 6 hours. Interestingly, ribosomal proteins showed up regulation even from 15 minutes, suggesting that cysteine induced protein translation inhibition may be the initial event, followed by imbalance in the levels of amino acids. This has now been mentioned in the discussion section (page 9) as data not shown.

Comment: Fig 4A, the growth inhibition of wild type strain is decreasing even in excessive cysteine conditions and seems to be normal after 30 hours. How can the authors explain this recovery?

Response: The recovery in the growth inhibition indicates that cells have response mechanisms that lead to cellular adaptation. This adaptation probably depends on Nc11, as its deletion abrogates this recovery at the later time point.

Comment: Fig 4C, it is hard to detect that either transcription or translation is significantly affected in ncl1Δ. Can the authors fill the dots with the same color for the same gene in both graphs? In addition, the authors should show mRNA and protein levels for both genotypes in Cys+Leu/Cys. Is the decoupling between transcription and translation resulting from the known NCL1 function?

Response: We understand the concern of the reviewer; however, in the scatter-plot which is just highlighting the correlation between different conditions, it will be difficult to show the profile of each gene across two conditions even with different colors. However, for the simplicity we have given the expression of each gene at the RNA and the protein level in the table format in the supplementary table 5.

To understand if the role of NCL1 is at transcription or translation, RNA-seq for both genotypes was performed only in cysteine treated condition. However, currently due to the paucity of resources, we could not perform global mRNA expression with Leu and Cys cotreatment in WT and ncl1Δ. Since, we had done proteomics for all the conditions and we know that branched chain amino acid genes are altered, we performed RT-PCR for 6 genes involved in branched chain synthesis and found that cysteine upregulates the genes involved in branch chain amino acids in both wild type and Δncl1. Protein levels of these genes were upregulated only in WT, but not in Δncl1, again highlighting the decoupling between transcription and translation. Since we have not been able to perform the mRNA expression at global levels, we have not included it in the revised manuscript. However, if the reviewer feels it is important we can include it.

The exact role of NCL1 in translation modulation is unknown, however, it has been shown to have an important role in the TTG codon mediated selective translation of proteins during oxidative stress (Chan, Pang et al. 2012). We have now included this statement in the revised manuscript in the discussion section (page 10).

Comment: Fig 5B, can the authors show relative leucine levels for both cells under excessive cysteine only condition?

Response: As suggested by the reviewer we have now replaced figure 5B with a new figure showing relative leucine levels for both cells under excess cysteine, leucine and leucine+cysteine.

Minor concerns:

Comment: Matrices in Fig 1B and 1E can lead to misunderstanding as if blue is negative, and red is positive. Since the growth percentage is between 0 and 100 ($+\alpha$), the color of the matrix should be changed to a linear gradient with a single color.

Response: We completely agree with the reviewer and have now altered the color gradient.

Comment: In Fig 4B, the color of the 1st box and 4th box is the same. In Fig 4D, the shape of the bar for 'branched-chain amino acid biosynthetic process' is different from others.

Response: We sincerely apologize for this and have now corrected it in the revised manuscript

Comment: In Fig 5E, what is Batp above the arrows?

Response: This was meant for Bat1 protein. We have now changed it in the revised manuscript

Comment: In Fig 5G, what does 'fold change' mean in y-axis?

Response: We apologize for this. We have now clearly indicated it in the y-axis of the revised manuscript

Reviewer 2

Comment: Bhat and co-authors present an analysis of the budding yeast metabolism in response to elevated cysteine levels in the medium. They show that leucine can partly abrogate the toxic effect of cysteine and they show that this rescue depends on Ncl1. Further, they show that the leucine metabolite KIC can rescue cysteine toxicity and that pyruvate down regulation in response to elevated cysteine levels is also involved in the toxicity.

This study is somewhat restricted because it focuses on the toxicity of a single amino acid (cysteine). However, it is scientifically sound and it is characterized by the application of a wide range of omics methods for the characterization of the cellular response to cysteine. The study

presents data about a whole-genome screen for interactors mediating the leucine rescue effect, further proteomics data, transcriptomics data, metabolomics data, polysome profiling and various other assays. This is a pretty extensive study.

Response: We thank the reviewer for finding the study scientifically sound.

Comment: I only have one major comment: the language has to be substantially improved. There are many grammatical errors in the manuscript, which definitely has to be revised by a native speaker. Further, some descriptions and figure labels are unclear (see below). It is puzzling that one of the authors' name was misspelled in the submission system. One would hope that the rest of the analysis was performed with greater care.

Response: we sincerely apologize for this. We have now made necessary corrections and hope that the reviewer finds them adequate.

Comment: Already at the beginning of the results section (page 3) it should be made clear which base-medium was used and how much higher than normal the AA concentrations were. When concentrations of AAs are given: do they refer to the amount added to the medium? Or do they refer to the total amount, i.e. what's already in the medium plus the added amount? Be more explicit and clearer.

Response: We thank the reviewer for raising this point. We have used synthetic complete media for this experiment which mainly contain Dextrose, Yeast nitrogenous bases and amino acid mixtures. We have now mentioned it in the beginning of the manuscript in page 3 at the relevant positions. We have used 10 fold molar excess of each amino acid compared to that present in the media. The amino acid concentration referred to is the amount added to the media. The concentration of each amino acid added is given in the supplementary table (S1).

Comment: Figure 1B: the color scale is confusing. When two colors are used they usually represent a reduction compared to normal and an increase compared to normal. Here, grey is not neutral, but some middle range growth, blue is low and red fast growth. That is counterintuitive.

Response: We have now corrected this as mentioned above in response to a similar comment by reviewer 1.

Comment: Page 4, proteomics experiments: which strain was used? Which cysteine concentration? How does the concentration compare to normal (how much greater than normal)?

Response: In page 4, the proteomics experiment was performed in Wildtype BY4741 strain. The concentration used for this experiment was 1mM. The concentration 1mM used in this experiment increases the intracellular concentration by approximately 4 fold. We intentionally used this moderate

concentration of cysteine and not a very high concentration like 5 mM, where other factors also could have an effect. Besides 1mM cysteine is considered to be pathophysiological.

Comment: Page 5, genomic screen: provide more details on the screen in the results section. More is given in the methods part, but you should also give more details here. How many genes were tested? What was the readout for the assay? I.e. what exactly was tested?

Response: As desired by the reviewer, we have now included more details about the gene screen in the result section (page 5)

Comment: Figure 3: How many strains (genes) were selected in each phase? Were the p-value corrected for multiple testing?

Response: We have now mentioned this in detail (page 5). We considered only genes that were significant. However, multiple correction was not done.

Comment: Still Figure 3A: The color red is used for two different things in the same figure that is very confusing. In the middle panel it indicates resistant strains, in the bottom panel sensitive strains. Further, the labels are too small.

Response: We thank the reviewer for pointing this out. We have now corrected it. The colour red is for sensitive strains. We have also magnified the labels.

7. Figure 4B and Suppl. Fig. 3B: Which proteins are shown here? I assume only proteins showing an effect to cysteine alone are shown. But this needs to be made explicit. Based on which statistical criteria were the proteins selected?

Response: We have selected the proteins that were differentially expressed in presence of cysteine and altered by leucine. We have calculated the p-value between control and cysteine treated cells to determine the differentially altered protein. The relative expression of these differentially expressed proteins was then compared with other conditions. For the selection of all of the proteins we have applied paired t-test and while calculating the p value, the fold change was log transformed.

Comment: Page 7, end of results: Leucine effects and pyruvate effects at equal concentrations are compared. Can one directly compare the molar concentrations between these two substances?

Response: We did not compare the molar concentrations between the two substances. We simply mentioned the extent of growth inhibition by 5 mM pyruvate and 5 mM leucine. We did it as the effect of pyruvate was maximum at 5mM and the concentration of leucine in the 10x amino acid media is approximately 5mM, and above this concentration, there is not much effect of leucine.

Comment Page 9: spell out 'CAD'.

Response: We thank the reviewer for pointing this out. We have now corrected this.

Comment Page 13, proteomics analysis: differential expression of proteins is defined based on fold-changes only, but no proper statistics is performed. I think the authors should at least log-transform the intensities and then perform a t-test. Likely this will not change the conclusions much, but it is the proper way to do this.

Response: We thank the reviewer for raising this point. We have performed an iTRAQ based relative quantification in order to identify the differentially altered proteins. As suggested by the reviewer we have log transformed the fold change and then performed statistical analysis (t-test). The data is now updated and shown in the supplementary table S5 and S6. In the table all the differentially expressed proteins, irrespective of their significance, are shown. However, for GO analysis we only considered proteins that were significantly differentially expressed.

July 22, 2019

RE: Life Science Alliance Manuscript #LSA-2019-00360-TR

Dr. Shantanu Sengupta
Institute of Genomics and Integrative Biology
Proteomics and Structural Biology Unit
Mall Road
Delhi, Delhi 110007
India

Dear Dr. Sengupta,

Thank you for submitting your revised manuscript entitled "Ncl1 mediated metabolic rewiring is critical during metabolic stress". As you will see, the reviewer appreciates the introduced changes and we would thus be happy to publish your paper in Life Science Alliance pending final revisions necessary to meet our formatting guidelines:

- please list 10 authors et al in your reference list
- please provide table S7 and S8 as word docx or excel files
- please add a callout in the text to FigS1E
- please mention the number of replicates for each figure and describe error bars depicted in the figure legends, name the statistical tests performed and add information on p-values (the latter where missing)
- please remove sentences from figure panels, these can go into the figure legends
- panels of Fig 2D are too small to be useful to the reader, please amend (there is also a spelling mistake in 'autoradiogram')
- please deposit the raw MS files in a repository (currently mentioned as being available upon request) and add the accession code to your manuscript text

A. FINAL FILES:

B. MANUSCRIPT ORGANIZATION AND FORMATTING:

Sincerely,

Andrea Leibfried, PhD
Executive Editor
Life Science Alliance
Meyerhofstr. 1

69117 Heidelberg, Germany
t +49 6221 8891 502
e a.leibfried@life-science-alliance.org
www.life-science-alliance.org

Reviewer #1 (Comments to the Authors (Required)):

The authors satisfactorily addressed all the concerns raised and the revised manuscript looks much improved. Since LSA journal allows only one round revision, I will not raise any additional comments at this time.

August 5, 2019

RE: Life Science Alliance Manuscript #LSA-2019-00360-TRR

Dr. Shantanu Sengupta
Institute of Genomics and Integrative Biology
Proteomics and Structural Biology Unit
Mall Road
Delhi, Delhi 110007
India

Dear Dr. Sengupta,

Thank you for submitting your Research Article entitled "Ncl1 mediated metabolic rewiring is critical during metabolic stress". It is a pleasure to let you know that your manuscript is now accepted for publication in Life Science Alliance. Congratulations on this interesting work.

DISTRIBUTION OF MATERIALS:

Again, congratulations on a very nice paper. I hope you found the review process to be constructive and are pleased with how the manuscript was handled editorially. We look forward to future exciting submissions from your lab.

Sincerely,
